# Does Informal Education Training Increase Awareness of Anaphylaxis among Students of Medicine? Before-After Survey Study

**DOI:** 10.3390/ijerph18158150

**Published:** 2021-08-01

**Authors:** Julia Leszkowicz, Agata Pieńkowska, Wojciech Nazar, Eliza Bogdan, Natalia Kwaka, Agnieszka Szlagatys-Sidorkiewicz, Katarzyna Plata-Nazar

**Affiliations:** 1Faculty of Medicine, Medical University of Gdańsk, Marii Skłodowskiej-Curie 3a, 80-210 Gdańsk, Poland; julia.leszkowicz@gmail.com (J.L.); apienk@gumed.edu.pl (A.P.); ebogdan@gumed.edu.pl (E.B.); natalia.kwaka@gumed.edu.pl (N.K.); 2Department of Pediatrics, Gastroenterology, Allergology and Nutrition, Faculty of Medicine, Medical University of Gdańsk, Nowe Ogrody 1-6, 80-803 Gdańsk, Poland; agnieszka.szlagatys-sidorkiewicz@gumed.edu.pl (A.S.-S.); katarzyna.plata-nazar@gumed.edu.pl (K.P.-N.)

**Keywords:** anaphylaxis, allergy, adrenaline auto-injector, survey, medical students

## Abstract

Allergies are among the most common chronic diseases in Europe. The most serious complication is anaphylactic shock. Most cases occur outside the hospital; thus, knowledge of symptoms and first aid is crucial. This study aimed to evaluate the awareness of anaphylaxis and the ability to use adrenaline auto-injectors among medical students, and to determine an improvement after training based on non-formal education. The research was conducted among 364 medicine students (years 1–5) from the Medical University of Gdańsk, with year-specific curriculum-based general medical knowledge. Training consisted of pre-test, practical training and post-test. Descriptive statistics were used to reveal the characteristics of students from different grades. A Mann–Whitney U test was used for statistical analysis. The tested students did not have sufficient knowledge to provide first aid in cases of anaphylaxis before training. There was an increase in knowledge (on average, 28.6%, *p* = 0.005) after training. Almost all (99.4%) of the respondents believed that they would be able to use an adrenaline auto-injector in case of emergency after the training. The training based on non-formal education was effective. The use of the subject-performed task method helped students to remember the stages of action in stressful situations.

## 1. Introduction

Over 120 thousand people in Poland suffer from anaphylaxis, and almost 100 of them die every year as a result [1]. European data reveal that incidence rates for all-cause anaphylaxis ranges from 1.5 to 7.9 per 100,000 people/year, with an estimate that 0.3% (95% CI 0.1–0.5) of the population will experience anaphylaxis at some point during their lifetime [2]. Most cases of anaphylactic shock happen outside the hospital, and this is why knowledge of symptoms and first aid is crucial. Adrenaline injected intramuscularly is the most effective treatment for this serious condition. [3,4]. Those who witness anaphylactic shock are often not prepared enough to react or call emergency services. As fast response time is crucial in this condition, many patients do not receive help on time [5].

Pediatric Scientific Circle from the Medical University of Gdańsk developed a project, in which the aim was to educate students of medicine about anaphylactic shock, and to show them how to use adrenaline auto-injectors (peer education). The project was based on non-formal education, with a subject-performed task method. Non-formal education relies on committed volunteers (in this case medical students), who firstly acquire knowledge about a given topic, then conscientiously prepare training programs and share knowledge with their peers later. The advantage of this method is its accessibility, as training can take place anywhere. Furthermore, the prepared topics are based on the needs of the target group and not on the overall curriculum, which often does not take into account the interests of the individual [6]. Moreover, projects like this provide a mutual benefit for both the students and the educators.

Similar pieces of training have been conducted among teachers [7], childcare personnel [8], school nurses [9,10], unlicensed assistive personnel [11] and medical students [12,13] in other countries, but not among medical students in Poland. This is the reason why we decided that knowledge of this topic should be ameliorated in this specific group.

We assumed that medical students were potentially one of the best prepared groups to act properly in similar emergencies. We hypothesized that medical students are not prepared well enough to manage anaphylactic shock properly. Short training based on the SPT method could improve the ability to recognize and act in the case of anaphylaxis.

This study aimed to evaluate the awareness of anaphylaxis and the ability to use adrenaline auto-injectors among medical students, as well as to determine an improvement after training based on non-formal education.

### Introduction to the Topic

Allergies are not only the most common chronic diseases in Europe, but also affect an increasing number of patients [5]. The most serious complication of an allergy is anaphylactic shock, which is a life-threatening condition. Anaphylaxis is a severe, dynamically progressing, systemic hypersensitive reaction to a given allergen [14]. The most common stimuli that cause anaphylactic shock are Hymenoptera venom, food and medicines [15,16]. The number of incidents of anaphylaxis has increased by 615% over the past 20 years [15,16]. Anaphylaxis typically presents as many different symptoms over minutes or hours. The most common affected areas include the skin (80–90%); respiratory (70%); gastrointestinal, containing: abdominal pain, diarrhoea, vomiting (30–45%); heart and vasculature (10–45%); and central nervous system (10–15%). There are usually two or more systems involved [4]. Anaphylaxis is diagnosed with a classification based on an individual’s signs and symptoms [14].

Diagnostic criteria include:Involvement of the skin or mucosal tissue plus either respiratory difficulty or low blood pressure.Two or more of the following symptoms after contact with an allergen:
Involvement of the skin or mucosa;Respiratory difficulties;Low blood pressure;Gastrointestinal symptoms.
Low blood pressure after exposure to a known allergen.

Infants and children—low systolic pressure (depending on age) or more than a 30% decrease in systolic pressure. Adults—systolic blood pressure <90 mmHg or more than a 30% decrease from baseline [14]. After the onset of symptoms, the most important thing is to terminate contact with the allergen, e.g., stopping the drug admission or removing the sting. The first-line drug is adrenaline (dilution 1:1000), which should be used at a dose of 0.5 mg in adults, and in children at 0.01 mg /kg body weight (max 0.3–0.5 mg). It is administered intramuscularly, preferably in the lateral surface of the quadriceps femoris muscle [3]. Research shows that anaphylactic shock occurs mostly in public spaces and at home. In such situations, adrenaline auto-injectors should be used, as they offer a fast and easy way to administrate the drug by a person without specific first aid training [17]. In Poland, an allergy sufferer must purchase a previously prescribed adrenaline auto-injector and carry it with them. This is the sole ‘source’ of adrenaline auto-injectors in both public spaces and at home [18].

## 2. Materials and Methods

The study consisted of three steps: asking respondents to fill in an anonymous questionnaire evaluating their initial knowledge regarding anaphylaxis; then, giving them a brief informative training about it; and, lastly, asking them to fill in the next anonymous questionnaire with the same questions (Appendix A) (Figure 1). The data were not paired. The questionnaire was designed by the students of medicine and members of the Pediatric Student Scientific Circle, and was based on a poster entitled “Anaphylaxis—an algorithm for emergency care”, published on the official website of the *World Allergy Organization* [19]. The bioethical commission gave its approval to carry out the study among medical students of Medical University of Gdańsk (no: NKBBN/148/2018). The research was conducted through April–May 2019 and involved 364 students from years 1 to 5 of the courses of schooling. Participants were recruited in small groups (10–20 people). The questionnaire consisted of 12 questions, divided into 3 sections (Appendix A).

The whole training took approximately one hour. We divided the time of training into a few sections: pre-test (10 min), theoretical and practical part of training (30 min), participants’ questions (10 min) and post-test (10 min). In the first part of the survey, the respondents were asked to specify the year of schooling, whether they had been diagnosed with allergies and if they had experienced an anaphylactic shock in the past. This part of the test was designed to divide the respondents into smaller groups and see if those variables correlated with the respondents’ knowledge of anaphylaxis. The next 3 questions evaluated general knowledge about anaphylactic shock, its symptoms and triggers. Further detailed questions concerned the treatment of anaphylactic shock, including drug doses and its routes of administration. In the end, respondents were asked about the correct first aid in case of anaphylactic shock, and how they assessed their ability to administer the treatment.

The second stage comprised of training on theoretical knowledge of anaphylaxis, and a practical part focused on learning how to properly use an adrenaline auto-injector. Ready-to-use trainer devices, provided by Viatris Inc. (Canonsburg, PA, USA), were used to simulate the use of a real auto-injector. The information provided in the training was based on the official World Allergy Organization Anaphylaxis Guidance [20,21].

The training scheme was always the same. In the beginning, the definition of an allergic shock and what can cause it was introduced. Then, the symptoms were discussed, concerning the respiratory, circulatory, digestive system and skin, and how each medication works as well as how to administrate it. The educators also talked about recovery positions and calling an ambulance. Because of the possibility of using the adrenaline auto-injector demonstrators, each participant could practice administration of the drug in case of shock, either on himself or colleagues. The SPT method was used, and action phrases were presented: “Grab the AAI: the blue part is on the top, the orange is down” “take off the protection from the top” “push the AAI on the external, lateral side of your thigh vigorously. Then repeat the action on your colleague”.

After the training, the respondents were re-tested, using a questionnaire with 13 questions. In total, 12 out of the 13 were identical to the first questionnaire, and the last question measured the respondents’ opinion about the usefulness of the training. Students used their phones to complete the survey that was created using the Google platform (Google Forms), and the results were automatically saved. This eliminated the human error in data entry, but created a difference between the number of surveys taken before and after the training. Unfortunately, due to technical difficulties, the training could not be carried out at the same time of day in each group, which could potentially affect the cognitive abilities and focus of the participants.

In this study, the principles of non-formal education were used: voluntary participation, training occurred after regular classes during free time; the method was adapted to the needs of the trainees; learning by practice; and partner relations with educators [6]. The main advantages of this method are: firstly, learning by practicing; and, secondly, deepening the knowledge in one field of science by repeating the same information every time by the educator. The educators also drew upon the subject-performed task method (SPT) [22]. The SPT method consists of educators giving very short commands that are accompanied by relatively extensive manual (non-verbal) training. Next, the commands and movements are repeated by students using a tangible item. The general procedure is that subjects are presented with verbal commands (e.g., “roll the ball”) and are asked to perform the action indicated by each command. Subsequently, the subjects are instructed to recall the commands presented. In the typical control condition, the subjects are presented with the same verbal commands as in the SPT condition, but without any instruction to act [23]. According to previous projects, students will memorize better, to a large extent, if the information is provided non-verbally [24]. Using an item (in this case AAI) during training makes it easier to remember subsequent stages of action, even under stress. Research on memory has shown that when a subject-performed task is compared with a traditional verbal task, the enactment of the task yields better memory performance than no enactment [25]. The SPT method has proven to be effective with children and adults [26]. This method was successfully applied in education [27], as well as medicine [28].

Mean and cohort-specific percentages of correct answers for each survey question were calculated. In addition to this, mean, median, mode and its frequency, minimum and maximum test score for each group of students classified by year of study were analyzed. Moreover, the answers of the respondents were explored in the following question-based groups: “I have an allergy to something: yes/no” and “I have seen anaphylaxis before: yes/no”. Due to the application of anonymous non-paired questionnaires, it was not possible to use statistical tests that analyze differences between a paired set of samples. Furthermore, the data are not normally distributed. Therefore, the Mann–Whitney U test was utilized to test for pre-post differences. The threshold of statistical significance was set at *p* ≤ 0.05.

## 3. Results

We collected 709 non-paired questionnaires, with 364 pre-tests and 345 post-tests (Table 1). Women comprised 60% of the sample. For test questions (4–11), the test score increased from pre-test 4.96 to post-test 7.25, out of 8 (Table 1). The mean increase in the test score for the whole sample was equal to 2.29 (*p* ≤ 0.05), and a 28.6% mean increase in the percentage of correct answers was observed.

Before the training, questions (Q) that had the highest percentage of correct answers were Q10 (94.0%), Q5 (82.7%) and Q6 (73.4%) (Table 2). After practical exercises, Q10 (97.7%), Q9 (96.5%) and Q7 (95.7%) were the best answered questions. The greatest increase was noticed for Q11 (73.1%), Q8 (60.3%) and Q9 (34.2%).

The greatest improvement in the mean test score was achieved by the students in the 1st (2.6, *p* ≤ 0.05) and 2nd (2.5, *p* ≤ 0.05) years, followed by the 4th (2.2, *p* ≤ 0.05), 3rd (2.1, *p* ≤ 0.05) and 5th (1.7, *p* ≤ 0.05) year students. Furthermore, after the training, no one scored less than four. In addition to this, the test score medians, as well as modes and their frequencies, increased in all year-of-study dependent cohorts.

Moreover, the percentage of positive answers to Q12 increased from 66.2% to 100.0% after training. In addition to this, 98.6% of the respondents answered “yes” to Q13, asked only after the training (Table 2).

When comparing students who had seen anaphylaxis (32 of 364, 8.8%) with those who had not, the pre-training test scores obtained by students who had seen anaphylaxis were lower at 4.53 than the scores of students who had not seen anaphylaxis at 5.01 (*p* < 0.05). After the training, this relation changed. The training scores were 7.30 and 6.76, respectively (*p* < 0.05) (Table 3).

No statistically significant differences in total scores of correct answers between allergic and non-allergic students were observed, both before (5.00, 4.89) and after the training (7.28, 7.19), respectively (Table 4). Nevertheless, an increase in total scores was observed.

## 4. Discussion

Before the training, total test scores were relatively low. After the training, students knew significantly more about anaphylactic shock (Table 1 and Table 2), and each trainee obtained a valuable minimum of at least 50% of the total test score at post-test. Collected data reveal that the training provided during the project was effective. Thus, our hypotheses that medical students are not prepared well enough to manage anaphylactic shock properly and that short training based on the SPT method could improve the ability to recognize anaphylaxis were confirmed. These results correspond with other studies, in which respondents also obtained very good results after receiving training in the management of food allergy and anaphylaxis. In every case, knowledge improved after a simple training [7,29,30,31]. Conducted research has also proved that specific multidisciplinary training increases self-efficacy [30] and confidence [7].

### 4.1. Emphasis on Practice

Before the training, questions with the highest rate of correct answers concerned the general knowledge of anaphylactic shock. After the training, the questions with the highest proportion of correct answers were the ones concerning the treatment (medication and routes of administration) (Table 1). Similar results were reported elsewhere [32]. When looking at specific question results, the biggest improvement was seen for Q8, Q9 and Q11 (Table 1 and Table 2), which referred to practical aspects of administering adrenaline by AAI. Moreover, there was nearly a 100% rate of correct answers to the question: “do you think the training was useful?”. This indicates that the program was efficient, and it emphasizes the need for training [7,29]. All respondents correctly answered at least 50% of the test questions after the training. Furthermore, almost 100% of respondents stated they would be able to use AAI, which is the most successful aspect of the project.

### 4.2. Awareness

In total, 33% of the study population reported being diagnosed with allergies. Based on other studies, 40% of the members of the general Polish population state that they suffer from allergies [33]. The level of knowledge about anaphylactic shock among allergy sufferers was not significantly higher than among non-allergy sufferers before the training, even though this group is more susceptible to anaphylactic shock than non-allergy sufferers (Table 4). There were also no statistically significant differences in total scores of correct answers between these two groups after the training. However, people who had seen an anaphylactic shock achieved significantly higher scores than people who had not seen an anaphylactic shock (Table 3). This indicates that witnesses of anaphylactic shock know that this is a dangerous situation, and shows a willingness to acquire knowledge on how to provide help [30]. On the other hand, the relatively large discrepancy in improvement may be biased, due to the ten times smaller number of participants who have seen anaphylaxis, compared to those who have not. Therefore, there is a need to show these results on larger groups, with a similar count of subjects in both control and experimental cohorts.

### 4.3. Discrepancy in Improvement

The results demonstrate that first and second year students achieved the greatest improvements in the test score after training. The initial years do not provide many practical classes, rather more theoretical and general knowledge courses concerning anatomy and physiology. These results indicate that a short training might significantly improve the knowledge of people who are not undertaking medical training or are not healthcare providers. The students in the third, fourth and fifth year achieved smaller improvements compared to the first and second year students after the training. This is also implied by the learning program from the Medical University of Gdańsk, which delivers the subject of anaphylaxis during those years [34,35]. Thus, a higher improvement rate in younger students, who have a relatively smaller understanding of medicine in general, is expected. On the other hand, the total test scores of the third, fourth and fifth year students were higher than that of the first and second years after the training. Therefore, even though the improvement rate was lower, the overall general knowledge was higher. This is to be expected, as students who are in higher years of study possess a larger general knowledge.

### 4.4. Problem with Regulations

Anaphylactic shock occurs most commonly outside of the hospital (triggered by contact with food or venom) [16]. Therefore, it is highly probable that an incident may happen in a public facility, such as schools, universities, restaurants or homes. There is evidence that proved that patients who received an early dose of adrenaline before arrival to the emergency department were at a significantly lower risk of hospitalization than patients whose adrenaline treatment was delayed [36]. This is the reason why adrenaline should be more available in public places. There are no obligatory school or national programs in Poland to educate on the management of anaphylaxis, except for the courses taught at the medical universities and related institutions. As not much information about anaphylaxis and its treatment is provided at schools in Poland, it is not surprising that students who continue education at university (even if they study medicine) have poor initial knowledge of this topic. This implies insufficient knowledge on this subject and a lack of abilities to help in case of an emergency. This project was carried out to bring attention to this problem, and start a discussion on the lack of education regarding the early treatment of anaphylaxis. Special attention must be paid by school personnel that are in charge of children at high risk of food anaphylaxis. Emergency plans, including comprehensive guidelines, should be set. For instance, the French school population includes 11 million children. One of 1500 children benefits from a care management project, and thus far no lethal cases have been recorded at schools [37]. Similar research conducted worldwide is proof that guidelines related to anaphylaxis management and training in the school setting are imperative to minimize the risk of a fatal incident [38,39,40]. This concerns not only school personnel, but also students themselves.

### 4.5. Limitations

When interpreting the results of this study, some limitations have to be discussed. The sample is generally representative of a group of medical students in Poland. The qualified medical curriculum is the same for every medical university in Poland, as the government states it. However, we cannot declare that the practical aspects of adrenaline injection are treated uniformly in every university. Nevertheless, this study underlines the need to adjust the curriculum in order to learn to manage anaphylaxis more practically.

Every training session differed slightly from others because it was performed by a group that consisted of various educators. A single group of educators, that were trained or created a video together, would deliver more consistent training and also more reliable and valid research results.

Another problem which could not be eliminated was the delivery of the training at different times of the day. Some groups of students were trained in the morning before classes, another around 11 a.m. and another after classes around 5 p.m. That is why there could be differences in the level of concentration and fatigue, affecting learning in both theoretical knowledge and practical skills. Delivering all training at the same time of day would have eliminated this issue. However, establishing one time suitable for all participants as well as educators was an impractical task, due to varying schedules, and would have led to a significantly smaller number of participants. Not all questionnaire results were analyzed because not every survey was completed (about 2% in total). Such data were disregarded.

Moreover, this study could have been conducted at other universities in Poland, Europe and other continents, to acquire a broader overview of medical students’ knowledge of anaphylaxis.

In addition to this, the reliability of the student-designed questionnaire was not tested. In the future, it can be estimated using the test score reliability coefficient, for example, Cronbach’s alpha.

Lastly, it would be better to pair pre-test and post-test test scores, for instance, by assigning a specific number to every student and applying a statistical test that analyzes paired samples.

### 4.6. Advantages

Compared with other studies on the management of anaphylaxis, this project varied in a few aspects. Similar pieces of training were conducted among teachers [7], childcare personnel [8], school nurses [9,10], unlicensed assistive personnel [11] or medical students [12,13] in other countries, but not among medical students in Poland. The training sessions were delivered by physicians or psychologists and were longer, whereas in this case, self-educated students delivered less time-consuming training. Despite these differences, a similar increase in knowledge and awareness was observed. This suggests that training based on non-formal education is as effective as training conducted by professionals. Another positive aspect of our study is the use of the subject-performed task method, which helps students to remember subsequent stages of action in stressful situations.

### 4.7. Future Directions

It was planned to further survey the same group of students in the following academic year, to evaluate the retention of knowledge. However, due to the COVID-19 pandemic, this did not happen. Regarding the very positive feedback after the training, similar pieces of training are planned to be conducted at schools after the COVID-19 pandemic.

The Polish program of education at schools includes procedures and protocols in case of bites, but does not specifically concern anaphylaxis and the administration of adrenaline [41]. The results of the study indicate that it would be valuable to introduce similar training programs in schools to educate young people and teachers on anaphylaxis and its treatment. If adrenaline in the easy-to-use form of auto-injectors was introduced to every school and university or public facility (as a part of the first-aid kit), training in anaphylactic shock would be obligatory. This would also contribute to the propagation and increase in the awareness of those issues among the general public, and improve the safety of those at risk of anaphylactic shock.

## 5. Conclusions

The knowledge on anaphylaxis among medical students is insufficient, although this group should be prepared to deal with this kind of emergency. However, training based on non-formal education was effective. The SPT method made it possible to deliver training in any place and at any time, while, more importantly, the opportunity to practice the use of auto-injectors provided a unique way to learn.

Our study has demonstrated that acquiring the skills needed to use adrenaline auto-injectors is quick and relatively easy. This suggests that non-healthcare providers could acquire those vital life-saving skills fairly easily. Collected data prove that performed training can significantly improve knowledge and practical skills.

## Figures and Tables

**Figure 1 ijerph-18-08150-f001:**
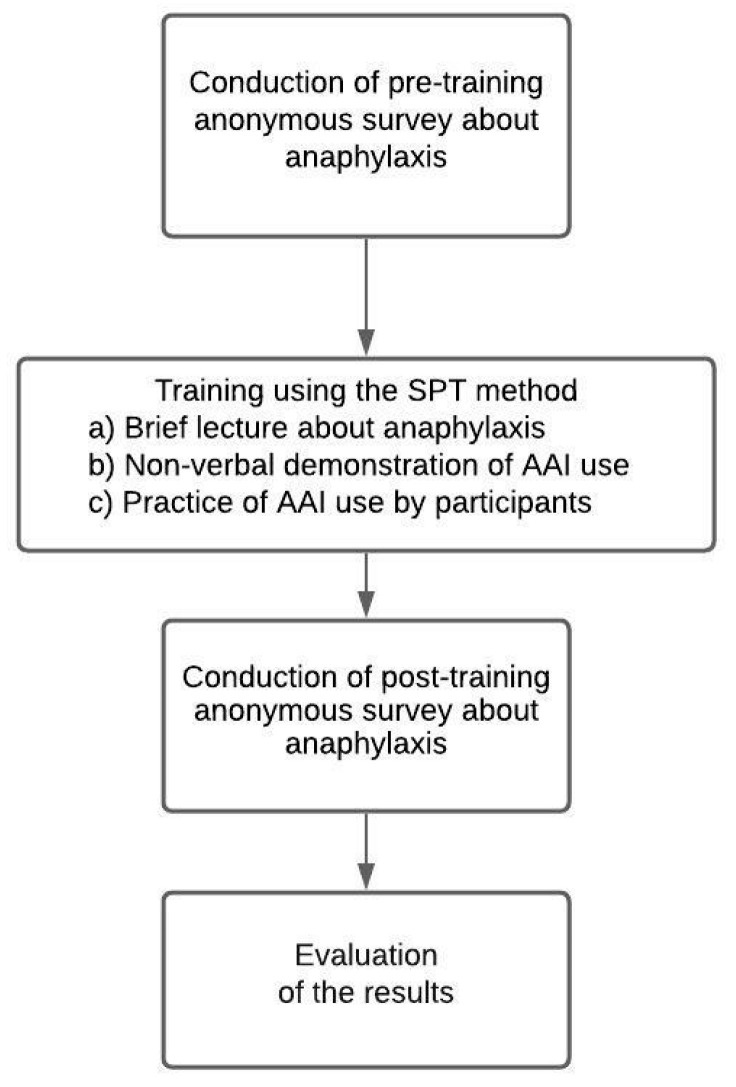
Study protocol.

**Table 1 ijerph-18-08150-t001:** Test scores by year of study, descriptive statistics.

**Pre-Test**									***p*-Value**
**Year of Study**	***n***	**Mean**	**Median**	**Mode**	**Frequency of the Mode**	**Minimum**	**Maximum**	**Standard Deviation**	
1st	93	4.06	4.00	4.00	28.00	1.00	8.00	1.52	-
2nd	108	4.71	5.00	4.00	28.00	1.00	7.00	1.37	-
3rd	58	5.50	6.00	6.00	26.00	2.00	7.00	0.98	-
4th	55	5.62	6.00	6.00	27.00	3.00	8.00	1.01	-
5th	50	5.84	6.00	6.00	20.00	3.00	8.00	1.00	-
Whole cohort	364	4.96	5.00	6.00	107.00	1.00	8.00	1.42	-
**Post-Test**									
**Year of Study**	***n***	**Mean**	**Median**	**Mode**	**Frequency of the Mode**	**Minimum**	**Maximum**	**Standard Deviation**	
1st	94	6.69	7.00	7.00	34.00	4.00	8.00	1.27	-
2nd	101	7.18	8.00	8.00	53.00	4.00	8.00	1.04	-
3rd	56	7.64	8.00	8.00	38.00	6.00	8.00	0.55	-
4th	48	7.79	8.00	8.00	42.00	5.00	8.00	0.62	-
5th	46	7.52	8.00	8.00	36.00	4.00	8.00	1.05	-
Whole cohort	345	7.25	8.00	8.00	198.00	4.00	8.00	1.08	-
**Pre-Post Test Difference**									
**Year of Study**	***n***	**Mean**	**Median**	**Mode**	**Frequency of the Mode**	**Minimum**	**Maximum**	**Standard Deviation**	
1st	1	2.63	3.00	3.00	6.00	3.00	0.00	−0.25	<0.001
2nd	−7	2.47	3.00	4.00	25.00	3.00	1.00	−0.33	<0.001
3rd	−2	2.14	2.00	2.00	12.00	4.00	1.00	−0.42	<0.001
4th	−7	2.17	2.00	2.00	15.00	2.00	0.00	−0.39	<0.001
5th	−4	1.68	2.00	2.00	16.00	1.00	0.00	0.05	<0.001
Whole cohort	−19	2.29	3.00	2.00	91.00	3.00	0.00	−0.35	<0.001

**Table 2 ijerph-18-08150-t002:** Pre-test and post-test percentages of correct answers to test questions.

Correct Answers to the Test Questions [%]	Question											*p*-Value
	Q4	Q5	Q6	Q7	Q8	Q9	Q10	Q11	Q12	Q13	Mean Q4–Q11	
Pre-test	77.8	82.7	73.4	67.9	26.4	62.4	94.0	12.1	66.2	-	62.1	-
Post-test	87.8	95.1	80.6	95.7	86.7	96.5	97.7	85.2	100.0	98.6	90.7	-
Pre-post test difference	10.1	12.4	7.2	27.8	60.3	34.2	3.7	73.1	33.8	-	28.6	0.005

**Table 3 ijerph-18-08150-t003:** Mean test scores by anaphylaxis related variables.

Criterion	Pre-Test *n*	Post-Test *n*	Mean Test Score Pre-Test	Mean Test Score Post-Test	Pre-Post Test Difference
Have seen anaphylaxis	32	29	5.01	6.76	1.75
Have not seen anaphylaxis	332	316	4.53	7.30	2.77
*p*-value	-	-	0.048	0.018	0.003

**Table 4 ijerph-18-08150-t004:** Mean test scores by allergy related variable.

Criterion	Pre-Test *n*	Post-Test *n*	Mean Test Score Pre-Test	Mean Test Score Post-Test	Pre-Post Test Difference
Allergic	118	111	5.00	7.28	2.28
Non-allergic	246	234	4.89	7.19	2.30
*p*-value	-	-	0.351	0.533	0.856

## Data Availability

The data presented in this study are available on request from the corresponding author.

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
