# Peer review of "Does Informal Education Training Increase Awareness of Anaphylaxis among Students of Medicine? Before-After Survey Study"

_ijerph, 2021, doi:10.3390/ijerph18158150_

Round 1

Reviewer 1 Report

Comments to "Does the informal education training increase awareness of anaphylaxis among students of medicine? Before-after survey study":

- Where is 1.1? Correct this section
- What are the starting hypotheses? We need to see if they are met or not once the methodology is applied
- Develops the review of previous literature
- Lines 51-54 need citation
- Includes a figure with the steps of the methodology
- Where has this method been successfully applied?
- The discussion of the results is very poor. Keep in mind that the work only has 9 pages including tables and references. Develop the results and discussions in relation to the hypotheses that you should include

Author Response

- Where is 1.1? Correct this section

//We changed into:

1.1.Introduction to the topic

- What are the starting hypotheses? We need to see if they are met or not once the methodology is applied

//We added:

“Our hypothesis is that medical students are not prepared well enough to manage anaphylactic shock properly.  Short training based on SPT method could improve ability to recognize anaphylaxis and act in case of it.”

- Develops the review of previous literature

//We added:

“Similar trainings were conducted among teachers [7], childcare personnel [8], school nurses [9,10], unlicensed assistive personnel [11] or medical students [12,13] in other countries, but not among medical students in Poland. This is the reason why we decided this topic should be ameliorated in this specific group.”

- Lines 51-54 need citation

//we deleted the following fragment:

“There was an assumption that medical students were potentially one of the best prepared group to act properly in similar emergencies. Despite that fact, most of them had never had an opportunity to perform adrenaline injection. Even among medical students the training turned out to be necessary.”

And we added a sentence:

“We made an assumption that medical students were potentially one of the best prepared group to act properly in similar emergencies.”

- Includes a figure with the steps of the methodology

//We added Fig.1

- Where has this method been successfully applied?

//we added a sentence:

This method was successfully applied in education [27] as well as medicine [28].

- The discussion of the results is very poor. Keep in mind that the work only has 9 pages including tables and references. Develop the results and discussions in relation to the hypotheses that you should include

//We changed the initial part of the discussion:

“The aim of this study was to evaluate the awareness about anaphylaxis and ability to use adrenaline auto-injectors among medical students as well as to determine an improvement after a training based on non-formal education.

Before the training obtained total test score were relatively low. After the training students knew significantly more about anaphylactic shock (Tab.I, Tab.II) and each trainee obtained the minimum of 50% score in post-test. Collected data reveals that the training provided during the project was effective. Thus, our hypothesis that medical students are not prepared well enough to manage anaphylactic shock properly and that short training based on SPT method could improve ability to recognize anaphylaxis were confirmed. These results correspond with other studies, in which respondents also obtained very good results after receiving training in the management of food allergy and anaphylaxis [7,24,25,26].”

Moreover, additional comments were added in sections 4.2, 4.3, 4.5 and 4.7 of the discussion.

The “results” section was also restructured, exact p-values were added.

Reviewer 2 Report

Dear authors,

Thank you for the opportunity to review the paper.

Interesting study.

This paper was to evaluate the awareness of anaphylaxis and ability to use adrenaline auto-injectors among medical students and to determine an improvement after a training based on non-formal education.

Pretest-posttest designs are widely used in behavioral research, primarily for the purpose of comparing groups and/or measuring change resulting from experimental education.

What is new? The knowledge about anaphylactic shock  among medical students is limited and improved significantly after clinical postings.

The method and material used is adequate. Also the discussion and the bibliography.

But in the results I believe the authors should explicated that:

The average pretest score was XX and that of the posttest was XX The percentage of improvement in mean score from pretest to posttest was XX.

Also I believe should Fischer's exact test was applied to analyze the improvement in scores between pretest and posttests.

I hope my comments may be of help to authors in their work.

Author Response

Interesting study.

This paper was to evaluate the awareness of anaphylaxis and ability to use adrenaline auto-injectors among medical students and to determine an improvement after a training based on non-formal education.

Pretest-posttest designs are widely used in behavioral research, primarily for the purpose of comparing groups and/or measuring change resulting from experimental education.

What is new? The knowledge about anaphylactic shock among medical students is limited and improved significantly after clinical postings.

The method and material used is adequate. Also the discussion and the bibliography.

But in the results I believe the authors should explicated that:

The average pretest score was XX and that of the posttest was XX The percentage of improvement in mean score from pretest to posttest was XX.

//At the beginning of the “Results” we added the following sentence:

For test questions (4-11), the score increased from pre-test 4.96 to post-test 7.25, out of 8 (Tab.I). The mean increase in the test score for the whole sample was equal to 2.29 (p ≤ 0.05) as well as a mean 28.6% increase in the percentage of correct answers was observed.

Also I believe should Fischer's exact test was applied to analyze the improvement in scores between pretest and posttests.

//Dear Reviewer, thank you for your suggestion. We did not applied Fischer’s test, as we have used Mann-Whitney U test to test for differences between test scores of our studied groups. Based on this data – p value – we can conclude that eg. post test scores where significantly higher than the pre test scores, and therefore the training of the students was useful. Thus, to the best of our knowledge, we do not need to use 2x2 tables that are used in Fischer’s test, as Mann-Whitney U test in (our) case of non-zero-one test scores is more appropriate than the Fischer’s test.

I hope my comments may be of help to authors in their work.

// Dear Reviewer, thank you for your all your comments.

Reviewer 3 Report

The topic is of importance and interesting.

I have some questions/suggestions for authors.

Line 47, I am surprised that the topic is not included in a qualified medical curriculum.

Line 57, "education" to "education."

Line 91, Line 154, anonymous questionnaire makes paired t-test impossible. Please mention this in "Material and Methods". And please discuss how to improve the study design if possible.

More description about sampling procedures needs to be included. 

Line 151, why "not tabulated"? 

Line 164, Line 222, the performance improvement is less in 5th-year students comparing to lower grade students. The explanation should be richer. Is this a problem? Should it be improved?

Line 171, Table II, why P-value is not provided for the comparison? Please add.

Line 178, Table III, why P-value is not provided for the comparisons? Please add.

Table III seems a little odd. Please refine it and add P-value.

Line 208-211, pre-test result showed that people who had seen anaphylactic shock case scored higher. however, the result was reversed in post-test. Need to explain this.

Line 241, the limitation should also mention the sampling process. Is the sample representative to any larger group, such as the medical students in Poland? Can this study inform any improvement in the curriculum design? If yes/no, should the limitation affect that?

Line 254, please provide the percentage here.

Line 272, the inclusion should be shortened to a half size if possible. It should address the most important results and implication of the study. 

Author Response

The topic is of importance and interesting.

I have some questions/suggestions for authors.

// Dear Reviewer, thank you for your all your comments.

Line 47, I am surprised that the topic is not included in a qualified medical curriculum.

//There is a lot of information about anaphylaxis in a qualified medical curriculum, but there is no practical training – only theoretical aspects. This is the reason why we wanted to tackle this subject. We strongly believe that more experience (in this case- simulation) is the key to successful action in the future.

Line 57, "education" to "education."

Line 91, Line 154, anonymous questionnaire makes paired t-test impossible. Please mention this in "Material and Methods".

//we changed all calculations from t-test to Mann-Whitney U test, as the sample are 1. Not paired, 2. They do not meet normal distribution.

Moreover, at the end of the Material and Methods section we changed the sentence to:

“Due to application of anonymous non-paired questionnaires it was not possible to use statistical tests that analyse differences between paired set of samples. Therefore, the Mann–Whitney U test was utilized to test for pre-post differences.”

We mentioned it in 4.5 Limitations:

“Lastly, it would be better to pair pre-test and post-test test score, for instance by assigning a specific number to every student, and apply a statistical test that analyses paired samples.”

More description about sampling procedures needs to be included.

//we added a flowchart (figure 1) and a sentence in “Materials and methods” section :

“Participants were recruited in small groups (10-20 people).”

Please discuss how to improve the study design if possible.

//we added a description in 4.5 Limitations:

“Lastly, it would be better to pair pre-test and post-test test score, for instance by assigning a specific number to every student, and apply a statistical test that analyses paired samples. Moreover, this study could have been conducted at other Universities in Poland, Europe and other continents, to acquire a broader overview of medical students’ knowledge of anaphylaxis.”

Line 151, why "not tabulated"?

//we did not included the p values in the tables and they are mentioned in the text only, therefore “not tabulated”. However, as we added p-values we deleted this sentence.

Line 164, Line 222, the performance improvement is less in 5th-year students comparing to lower grade students. The explanation should be richer. Is this a problem? Should it be improved?

//we added a description in part 4.3 Discrepancy in improvement:

“ Thus, higher improvement rate of younger students, who have relatively smaller understanding of medicine at all, is awaited. On the other hand, the total test scores of the third, fourth and fifth where higher than the first and second year after the training. Therefore, even though the improvement rate was lower, the overall general knowledge was higher. This is to be expected, as students who are in higher years of study show posses larger general knowledge.

That might be the factor of knowledge retention comparing initial and subsequent years. "

Line 171, Table II, why P-value is not provided for the comparison? Please add.

// p-value was added.

Line 178, Table III, why P-value is not provided for the comparisons? Please add.

Table III seems a little odd. Please refine it and add P-value.

//we split Table 3 into table 3 and 4. We added p-values

Line 208-211, pre-test result showed that people who had seen anaphylactic shock case scored higher. however, the result was reversed in post-test. Need to explain this.

We added a sentence:

“On the other hand, a relatively large discrepancy in improvement maybe biased due to ten times smaller number of participants who have seen anaphylaxis compared to those who have not. Therefore, there is a need to prove this results on larger groups, with similar count of subjects in both control and experimental cohorts.”

Line 241, the limitation should also mention the sampling process. Is the sample representative to any larger group, such as the medical students in Poland? Can this study inform any improvement in the curriculum design? If yes/no, should the limitation affect that?

//We added:

This sample is representative to a group of medical students in Poland in general. The qualified medical curriculum is the same for every medical university in Poland, as the government states it. However, we cannot declare if the practical aspects of adrenaline injection are treated uniformly in every university. Nevertheless, this study underlines the need to adjust curriculum to learn managing anaphylaxis in a more practical way.

Line 254, please provide the percentage here.

We added „(about 2% in total)”, at the end of 4.5 Limitations sections

Line 272, the conclusion should be shortened to a half size if possible. It should address the most important results and implication of the study.

//we added new conclusions, the last paragraph of the previous conclusions was redesigned and added to future directions:

Conclusions:

“The knowledge about anaphylaxis among medical students is insufficient, despite the fact that this group should be prepared to deal with this kind of emergency. However, the training based on non-formal education was effective. Using the SPT method made it possible to deliver training in any place and at any time, while, more importantly, the opportunity to practice the use of auto-injectors provided a unique way to learn.

The survey shows that 33% of study population was at risk of anaphylactic shock, as they were allergy sufferers. Frequency of this chronic disease is undeniable and it is immensely important to understand the symptoms and know how to administer adrenaline quickly and effectively. Our study has demonstrated that acquiring skills needed to use adrenaline auto-injectors is quick and relatively easy. This means that non-healthcare providers could acquire those vital life-saving skills fairly easily.

Collected data proves that performed training can significantly improve knowledge and practical skills. This model of education might be successfully implemented in schools or other public facilities to help people overcome the fear in stressful situations like anaphylactic shock.”

Future directions:

“Polish program of education at schools includes procedures and protocols in case of bites but does not specifically concern anaphylaxis and administration of adrenaline [41]. However, schools or first-aid classes at universities focus on CPR in case of a cardiac arrest, which could be an effect of an anaphylactic shock. The results of the study indicate that it would be valuable to introduce similar training programs in schools to educate young people and teachers on anaphylaxis and its treatment. If adrenaline in easy-to-use form of auto-injectors was introduced to every school and university or public facility (as a part of first-aid kit), training about anaphylactic shock would be obligatory. This would also contribute to propagation and increase of the awareness of those issues among the general public and improve the safety of those at risk of anaphylactic shock. “

Round 2

Reviewer 1 Report

Dear authors,

The submitted manuscript has improved with the latest corrections. Congratulations